# The Effects of Behavioral Relaxation Training on Academic Task Completion Among Students with Autism in Inclusive Classrooms: A Single-Subject Design Study

**DOI:** 10.3390/bs15121633

**Published:** 2025-11-27

**Authors:** Yitong Jiang, Hongmei Liu, Yin Wang, Xiaoyi Hu

**Affiliations:** 1Department of Special Education and Clinical Sciences, University of Oregon, Eugene, OR 97405, USA; yitongj@uoregon.edu; 2School of Special Education, Education Research Center for Children with ASD, Faculty of Education, Beijing Normal University, No. 19 Xinjiekouwai Street, Beijing 100875, China; wangyin21@mail.bnu.edu.cn (Y.W.); huxiaoyi@bnu.edu.cn (X.H.)

**Keywords:** autism, inclusive education, behavioral relaxation training, academic performance, single-subject design

## Abstract

To help students with autism reach their potential, high-quality inclusive education emphasizes the importance of their full participation in school and classroom activities. Academic anxiety, however, can interfere with students’ ability to follow instructions and, thereby, negatively affect their performance. We used a concurrent multiple-probe across tasks design within a single-subject research framework to coach students with autism in inclusive settings on relaxation behaviors to reduce their tension and increase their ability to complete tasks in Chinese literacy, mathematics, and English. Results indicated that behavioral relaxation training (BRT) produced positive effects on both behavioral and physiological outcomes and significantly improved academic task completion. Behavior changes stayed at a high level during the maintenance phase. In interviews, both caregivers and students reported positive attitudes toward the intervention, expressed strong acceptance of procedures, and confirmed that relaxation strategies facilitated assignment completion. Study limitations and implications for future research and practice are discussed.

## 1. Introduction

Students with autism face challenges in social communication, behavior patterns, and development of interests ([2]), and sometimes need additional support for their social, emotional, and behavioral competencies. Considerable research has explored educational programs aimed at promoting developments in these areas. Students with autism may also face difficulties in various academic areas, including reading comprehension, writing, and math (e.g., [4]; [12]; [22]; [43]; [50]; [64]). Even for those without intellectual disability, whose autism symptoms may be milder ([13]; [61]), reading comprehension appears to be significantly delayed compared to that of peers, as does performance in mathematical reasoning tasks, inferential processing, and problem solving ([41]; [54]; [56]). Beyond cognitive issues, researchers have claimed that students with autism could experience academic challenges due to multiple other reasons, both environmental and individual, such as lack of support, heavily verbal instruction delivery, large class size, low teacher-to-student ratio, social communication difficulties, resistance to change, sensory hypersensitivity, and bullying (e.g., [5]; [36]; [44]; [45]). As growing numbers of students with autism enter the general education system, increasing attention is being directed toward identifying existing issues and investigating approaches for promoting academic achievement among these students ([17]), although the number of empirical studies remains limited. In China, the inclusion of students with autism in general or inclusive education settings has also grown in recent years. A key objective of high-quality inclusive education is to ensure these students can actively participate in diverse learning activities and achieve holistic development, meeting the needs and expectations of both the students and their community supporters, including families and teachers.

### 1.1. Academic Performance and Academic Anxiety

One common academic challenge for students with autism is completing assignments on time. Doing so is essential if a child is to keep pace with instruction, maintain active engagement in learning activities, and, ultimately, reach their full potential and maintain a sense of self-efficacy ([9]; [38]; [53]). An increasing number of studies are exploring effective approaches for supporting academic performance in the population of students with autism, and some have reported promising results. For example, critical strategies for reading comprehension include anaphoric questions, direct instruction, and reciprocal teaching ([19]; [66]). Regarding math, interventions for students with autism ranged from more general strategies, such as differential reinforcement, time delay, correction, modeling or video modeling, and task analysis, to more specific strategies, including number lines, cardboard clocks, counting on, and next-dollar techniques ([6]; [20]; [27]; [32]; [62]). The majority of academic achievement programs for students with autism, however, have focused on directly coaching students with specific subject learning strategies or prompting educators to use particular teaching strategies, while one relevant aspect has remained under-explored: mental health.

Though findings have been inconsistent, several studies have shown significantly negative associations between academic achievement and depression and anxiety ([11]; [35]; [39]; [51]). [72] ([72]) argued for a correlation with the elevated physiological arousal associated with anxiety, which could result in attention problem and affect the ability to concentrate on academic activities. [25] ([25]) also found that students’ elevated physiological stress responses during exams were significantly correlated with lower grades, suggesting that heightened stress, represented by physiological arousal, can impair academic performance. Others have suggested that internalizing and externalizing difficulties have a negative impact on adaptive functioning, academic self-efficacy, and academic performance ([37]; [42]). Though many of these earlier studies have included only typically developing students, the same issues appear among children with autism. In fact, [67] ([67]) found that children with autism are more likely to experience mental health difficulties compared to their typically developing peers. Studies have also indicated that anxiety and its impacts can be exacerbated in students with autism because of the sensory, social, and academic demands of school settings ([1]; [3]).

As symptoms of anxiety are often verbally mediated and communicated, targeting treatments such as cognitive-behavioral therapy (CBT) and Acceptance and Commitment Therapy (ACT) often include strategies that rely heavily on expressive and receptive language and that use affect labeling, coping self-talk, and cognitive restructuring ([15]; [29]). To date, multiple group-design studies have examined the relation between academic performance and academic anxiety, as well as addressed related challenges through various interventions ([68]; [70]). In addition, researchers have validated the effectiveness of existing treatments, such as ACT and CBT, for reducing academic anxiety, physiological arousal and anxiety-related disruptive behaviors, and for enhancing student performance, using single-subject research designs ([26]; [59]). However, notable gaps remain in both research and practice in this field. First, treatments like ACT and CBT can be complex and often require support from another individual (typically an adult) because they demand a high level of cognitive engagement. These strategies, therefore, can be challenging for students with autism who have low levels of cognitive and language skills. Thus, researchers have started to look at other behavioral analytic approaches that might be appropriate for individuals with autism who have particularly low developmental status. Studies using systematic desensitization, modeling, reinforcement, graduated exposure, stimulus fading, and safety signals to interrupt anxiety-related behaviors (e.g., crying, screaming, refusing, shaking, eye widening, running away, vocal resistance) have reported promising results ([28]; [55]; [58]; [71]). Most of these studies, however, focus on anxiety related to specific situations or items, such as dental examinations, activity rooms, needles, and toilet flushing. The results, therefore, may not apply to diverse contexts, including academic activities, and may not be generalizable. Second, rather than using direct assessments, previous studies often relied on participants’ self-reported physiological changes or on measures of disruptive behavior (e.g., [24]; [26]; [60]). The former approach can be challenging for individuals with autism, particularly those with language delays or difficulties in self-monitoring. The latter, meanwhile, may make it difficult to distinguish anxiety-related behaviors from other functions, such as attention-seeking or self-stimulatory behaviors. Finally, to our knowledge, there is a lack of evidence from single-subject designs examining the effectiveness of interventions on academic anxiety and academic performance while incorporating physiological measures in individuals with autism. Therefore, further investigations are needed to clarify the relationship between academic anxiety, relevant physiological indicators, and academic performance, as well as to identify potentially beneficial treatments for this population.

### 1.2. Behavioral Relaxation Treatment (BRT)

BRT, developed by [57] ([57]), is a relaxation program that arose out of applied behavioral analysis, which emphasizes the measurement of observable and demonstrable relaxation behaviors rather than subjective feelings of relaxation. [34] ([34]) defined BRT as a training program that teaches individuals to understand, master, and perform relaxation behaviors or actions across ten body parts, thereby achieving significant behavioral relaxation effects. These researchers developed a set of ten “relaxed behaviors” to be taught by verbal description, modeling, and physical prompting:

Head-supported by the chair cushion and straight with respect to body midline; eyes-lids closed smoothly and no motion of eyes beneath them; mouth-lips and teeth slightly parted; throat-motionless (i.e., no swallowing); shoulders-rounded, transecting the same horizontal plane, motionless; body-torso, hips, and legs symmetrical around midline, motionless; hands-resting on chair armrest or on thighs, palms down with fingers slightly curled; feet-heels on chair footrest with toes pointing in a V; quiet[note 1]-no vocalizations or loud respiratory sounds; breathing-slow and regular.([34])

One of the theoretical foundations of BRT is the four-domain response theory of complex behavior. This theory proposes that complex behaviors involve the co-variation of four response domains, including motor, verbal, instinctive, and observational, such that a response in one domain increases the likelihood of responses in the others ([34]). Within BRT, students with autism are guided to notice their overt relaxation behaviors, which can, in turn, elicit relaxation across other domains. This process is believed to facilitate comprehensive relaxation, encompassing both overt and covert responses in all four domains.

A growing body of evidence supports the effectiveness of BRT across multiple areas. Initially, BRT was most frequently applied in pain management. Studies have shown that BRT can reduce migraine symptoms, trigeminal neuralgia pain, and daily pain interference ratings while also decreasing reliance on analgesics and anti-anxiety medications ([8]; [18]; [40]). [65] ([65]) employed behavior skills training to teach relaxation behaviors to undergraduates who reported test anxiety and found that students in the experimental group demonstrated significantly lower anxiety scores compared to those in the control group. Researchers have also examined BRT among individuals with disabilities. [33] ([33]) documented significant reductions in behavioral anxiety and pulse rate in adults with intellectual disability. [21] ([21]) demonstrated the benefits of BRT for an adult with a brain injury, including reduced tremors and improved communication via letter board, particularly when BRT was paired with biofeedback.

The scope of BRT research also includes younger populations. Early research by [52] ([52]) revealed decreases in hyperactive behaviors among children diagnosed with attention-deficit/hyperactivity disorder (ADHD). [46] ([46]) showed that BRT reduced challenging behaviors in adolescents with intellectual disabilities, while [7] ([7]) observed significant improvements after using BRT with two students with autism (aged 19 and 29 years) who were experiencing test anxiety. The latter finding suggests BRT may have benefits for academic performance. However, none of these studies were conducted in school settings, nor did they sufficiently consider school-based applications (e.g., outcome selection and measurement). Moreover, they did not include younger students with autism, such as those in inclusive elementary or middle schools, who face real challenges in academic settings. Therefore, research on the impact of BRT on academic achievement in younger children with autism who demonstrate a need is significant for both research and practice.

### 1.3. The Present Study

We aimed in the present study to examine the relations among BRT, behavioral and physiological relaxation, and improvement in task completion in Chinese literacy, mathematics, and English for students with autism enrolled in general education schools in China. Specifically, our study addressed the following research questions:Do students with autism demonstrate increased relaxation following BRT?Does task completion in the three core subjects (Chinese literacy, mathematics, and English) improve after students with autism receive BRT?Are improvements in task completion maintained after the intervention ends?

## 2. Methods

### 2.1. Participants

Two students were recruited from inclusive public primary schools in China. Inclusion criteria required participants to (a) have a diagnosis of autism, (b) be enrolled in general education settings, (c) demonstrate basic academic skills but show inconsistent task completion, and (d) exhibit observable signs of tension or difficulty relaxing (e.g., gripping clothing, clenching the jaw), as reported by both teachers and parents. Children were excluded if they had previously received relaxation training, were currently participating in such services, or were taking medications likely to affect physiological arousal or relaxation response (e.g., anxiolytics, sedatives). Parental consent was obtained for both participants to take part in this study.

Bob (pseudonym) was a 14-year-old sixth-grade student who could follow multi-step verbal instructions and engage in reciprocal conversations. He demonstrated reading comprehension, sentence construction, and the ability to solve basic word problems. In English, he could name familiar objects and participate in simple conversational exchanges. During academic tasks, however, Bob frequently showed signs of distress (e.g., saying “Are you trying to set me up?”, rocking his chair, and repetitively scratching his head), and required frequent prompting to remain on task.

Sam (pseudonym) was an 8-year-old second-grade student who used complete sentences to express his needs and initiated brief conversations centered on preferred topics. He performed at grade level across core subjects and demonstrated ability in oral reading, spelling, single-digit multiplication, and basic English copying tasks. Despite these strengths, Sam exhibited anxiety-related behaviors during academic activities, such as frequently asking, “Is there a lot of homework today?”, along with motor tics and finger-biting. He often left his seat to interact with non-task materials and required repeated prompts to re-engage with academic tasks.

### 2.2. Setting

All sessions were conducted in a university-affiliated autism education research center. BRT and physiological tests were conducted in the reading area, while academic task probes were administered in the one-on-one instruction area (3 × 3 m) furnished with a standard school desk (1.6 m × 0.8 m) and chairs. Sessions were terminated if a participant demonstrated continuous emotional distress or challenging behaviors persisting for over five minutes.

### 2.3. Research Team

The intervention was delivered by a Board Certified Behavior Analyst (BCBA) and a graduate student in special education. Both interventionists had prior experience supporting children with autism and received targeted training specific to the Behavioral Relaxation Training (BRT) protocol. This training included modeling of the intervention procedures, followed by role-play practice to ensure procedural competence. The fourth author, a university professor with expertise in behavioral interventions, provided real-time supervision throughout the sessions to ensure correct and consistent implementation of the protocol.

### 2.4. Research Design

A concurrent multiple-probe design across academic tasks (Chinese literacy, mathematics, and English) was adopted. This design minimizes potential practice effects associated with repeated baseline measurement while allowing each participant to serve as their own control ([23]). Although only two participants were included, each participant experienced the intervention across three distinct academic tasks, enabling repeated demonstrations of the treatment effect and establishing functional relations. The design also provided flexibility for implementation under real-world school constraints, such as scheduling and instructional arrangements.

### 2.5. Intervention Materials

Two types of materials were used during the intervention: visual image cards and video models of relaxation behaviors.

The image cards consisted of twelve laminated color cards (12 cm × 12 cm) depicting a specific body part or relaxation-related action ([34]): trunk, buttocks, head, shoulders, hands, feet, throat, quiet, breathing, mouth, eyes, and eyelids. The cards served as visual aids during instruction to facilitate identification and practice of target relaxation behaviors.

The video models of relaxation behaviors comprised a series of short clips developed to visually demonstrate each target behavior corresponding to the audio guide. Each video featured a trained model performing the specific relaxation posture, accompanied by synchronized on-screen text labels (e.g., posture name displayed in black font on a white background) and a voiceover providing the behavioral definition.

### 2.6. Dependent Variables and Measurement

#### 2.6.1. Primary Dependent Variable

Academic task completion was defined as the percentage of items completed in each of three subjects (Chinese literacy, math, and English) within a 10-min session. Tasks were aligned with each participant’s grade level and, to minimize the effects of new learning, consisted of previously mastered content. During each session, the instructor (the third author) administered one worksheet per subject and recorded the proportion of attempted items out of the total assigned items. Notably, task accuracy was not recorded, as the primary focus was on behavioral engagement and sustained task participation rather than academic correctness. To ensure task equivalence across sessions, academic worksheets were selected based on parental reports of each participant’s recent learning content. All items were drawn from previously taught materials and reviewed to match the participant’s current academic level. Each worksheet contained a fixed number of items per subject and maintained consistent formats and item types across sessions. Task difficulty was kept stable both within and across subject areas by standardizing cognitive demand (e.g., similar linguistic structures in literacy tasks; comparable arithmetic complexity in math). This controlled approach minimized the influence of task variability on academic performance outcomes.

For Bob, Chinese literacy tasks included converting Pinyin to characters (e.g., converting “shūbāo” to “书包” [school bag]) and identifying antonyms (e.g., for “清晰” [clear], the correct antonym is “模糊” [vague]). Math tasks consisted of multi-step word problems and arithmetic operations, while English tasks required vocabulary translation (e.g., translating “apple” into “苹果”), multiple-choice comprehension questions, and sentence translation.

For Sam, Chinese literacy tasks consisted of converting Pinyin to characters, word formation (e.g., given the character “水” [water], acceptable answers included “水杯” [water cup] or “水果” [fruit]), and semantic pairing (e.g., matching “灌溉” [irrigate] with “农田” [farmland]). Math tasks included mixed addition, subtraction, and multiplication of numbers up to ten; English tasks involved copying vocabulary words.

#### 2.6.2. Secondary Dependent Variables

Behavioral relaxation was measured using the Behavioral Relaxation Scale (BRS; [49]), a standardized observational tool assessing ten observable behaviors indicative of relaxation. These directly correspond to the motor domain of the four-domain response theory, which emphasizes low skeletal muscle activity and symmetrical posture as external indicators of relaxed states. Each assessment involved a one-minute observation period. Breathing rate was recorded during the first 30 s. The other nine behaviors were observed over the next 15 s. During the final 15 s, each behavior was scored as “relaxed” or “not relaxed.” The proportion of behaviors coded as relaxed provided the relaxation score. A relaxation score of 90% or higher was used to classify participants as relaxed, reflecting high behavioral consistency with tolerance for minor variations.

Physiological relaxation was measured using a BIOPAC MP160 system, which recorded muscle tension via electromyography (EMG) and pulse rate via a photoplethysmographic (PPG) sensor. These indicators correspond to the instinctive domain, as defined by autonomic nervous system responses (e.g., heart rate, muscle tone). Physiological relaxation was defined as an EMG amplitude of less than 5 microvolts (μV) and a pulse rate below 90 beats per minute (bpm). Both behavioral and physiological measures were collected during pre- and post-intervention probes. This information provided data for assessing changes associated with the relaxation training.

In this study, the primary focus was on the motor and instinctive domains of the four-domain response theory, operationalized, respectively, through observable posture-based relaxation behaviors (via BRS) and autonomic indicators such as muscle tension and heart rate. The observational and verbal domains were not directly measured; however, they were implicitly involved in the intervention process. Specifically, participants were required to visually monitor and adjust their own posture to meet target behavioral criteria (engaging the observational domain) and followed verbal instructions or internalized cues (engaging the verbal domain) to support relaxation performance.

### 2.7. Experimental Arrangements and Procedures

#### 2.7.1. Pre-Intervention Probes

Each participant completed three five-minute probes prior to the intervention to assess initial behavioral and physiological relaxation. During each probe, participants reclined on a beanbag and were fitted with surface EMG and PPG sensors to monitor forehead muscle activity and pulse rate. The instructor said, “Please relax by yourself now. I will let you know when the time is up,” and initiated a five-minute timer. Participants remained on the beanbag and were allowed to adjust their posture as needed. The instructor observed quietly throughout the session and scored behavioral relaxation using the BRS, without offering any prompts or feedback. After each session, the probe administrator (the third author) and a graduate student assistant (the second author) extracted the physiological data, including EMG amplitudes and pulse rates, for analysis.

#### 2.7.2. Baseline

The instructor conducted baseline sessions twice weekly. Each session included three consecutive 10-min segments for Chinese literacy, three for math, and three for English tasks. Participants were provided with one worksheet for each subject, and were instructed to complete as many items as possible within the 10-min time limit. No prompts or feedback were provided. A minimum of five data points were collected for each participant.

#### 2.7.3. Prerequisite Skill Training

The prerequisite skill training phase consisted of three sequential components: (a) body part tacting, (b) individual relaxation behavior learning, and (c) integrated behavior skills training.

##### Body Part Tacting

A discrete trial teaching format was used to teach participants to tact body parts and relaxation-related actions. During each trial, the instructor presented a visual image card and asked, “What is this?” Participants were given three seconds to respond. Independent correct responses were immediately reinforced. If the response was incorrect or absent, the instructor verbally modeled the correct tact and re-presented the same card with the original question. Afterwards, the instructor probed participants’ skills by randomly presenting 12 cards without providing reinforcement or feedback. Mastery was defined as 100% correct independent responses across two consecutive probes.

##### Individual Relaxation Behavior Learning

Each relaxation behavior was taught individually using a behavior skills training approach comprising four core components: instruction, modeling, rehearsal, and feedback ([30]). The instructor first provided a clear description of both the relaxed and non-relaxed forms of the target posture (instruction), demonstrated the correct relaxed posture (modeling), then, using a directive (e.g., “Relax your [target body part]. Hold this position for 30 s until you hear my voice again.” [rehearsal]), prompted the participant to perform the behavior. A correct response was defined as independently initiating and maintaining the relaxed posture for the full 30 s. Correct responses were immediately reinforced with descriptive praise (feedback). If the participant responded incorrectly or failed to respond, a video model of the correct behavior was presented. The instructor offered continuous verbal or physical prompts until the participant exhibited correct responses. Mastery required the participant to demonstrate 100% correct independent responses across two consecutive probes accompanied by the same directive but without any prompting or feedback.

##### Integrated Behavior Skills Training

This component of prerequisite skills training integrated the ten relaxation behaviors into a complete response chain and comprised two sections: sequential tacting and whole relaxation behavior acquisition.

During sequential tacting, the instructor asked participants to verbally name all ten relaxation behaviors in a fixed order (body, head, shoulders, hands, feet, throat, quiet, breathing, mouth, and eyes). The instructor delivered social reinforcement for correct responses made within three seconds, or, following errors or non-responses after five seconds, modeled for the participants how to name the behaviors. Mastery was defined as 100% correct responses across two consecutive probes.

In whole relaxation behavior acquisition, participants practiced the entire relaxation sequence with guidance. Across ten trials, the instructor delivered stepwise instructions (e.g., “Relax your body.”) and allowed five seconds for an independent response. Correct responses were praised; incorrect or missing responses prompted a video model before the instructor moved to the next instruction.

During performance probes, the instructor directed participants to perform the full sequence of relaxation behaviors without prompting or feedback. Advancement to the intervention phase required at least 90% accuracy across two consecutive probes. Participants who did not meet this criterion repeated the whole relaxation behavior acquisition section for additional instruction.

#### 2.7.4. Intervention

For each intervention session, participants followed the procedures described above for whole relaxation behavior acquisition, then completed a 15-min BRT and finally engaged in a 10-min academic task probe. The 15-min BRT consisted of a 10-min guided relaxation practice with praise or prompt provided every 30 s, then a 5-min independent relaxation probe without feedback. Participants proceeded to the academic task probe only after being considered relaxed (i.e., having been coded as relaxed during at least 90% of the observed behaviors). Both of our participants required no more than two BRT sessions before starting the task probes. Academic task probes followed the same format as task probes in baseline sessions. When task completion for a given subject exceeded 80% across two consecutive probes, the intervention began for the next academic subject.

#### 2.7.5. Post-Intervention Probes

Participants’ relaxation performance across behavioral and physiological dimensions was evaluated during post-intervention probes in the same way it was evaluated in pre-intervention probes.

#### 2.7.6. Maintenance

Maintenance sessions were conducted after each participant had completed all intervention phases across the three academic subjects. These sessions occurred during the first, second, and fourth weeks following the final intervention session. Procedures were identical to those used during the baseline phase.

### 2.8. Procedural Fidelity

A graduate student majoring in special education was trained to assess procedural fidelity using a checklist developed specifically for this study. At least 30% of sessions from each phase were randomly selected for review. Procedural fidelity was calculated as the percentage of correctly implemented steps relative to the total number of steps. Results indicated 100% fidelity across all phases and participants, confirming that the intervention procedures were implemented as designed.

### 2.9. Interobserver Agreement (IOA)

A graduate student in special education was trained in the study’s operational definitions and data recording procedures. IOA was assessed for at least 30% of randomly selected sessions across participants and study phases using point-by-point agreement. IOA was calculated as agreements/(agreements + disagreements) × 100%. Results showed that IOA values consistently exceeded 97%, indicating high reliability of data collection.

### 2.10. Social Validity

At the conclusion of our study, we used caregiver questionnaires and participant interviews to assess social validity with Bob, Sam and their mothers. On a 13-item Likert-type scale, caregivers rated the intervention as highly acceptable and beneficial, and indicated that they noticed observable improvements in their children’s relaxation and academic task performance. Sam’s mother reported full satisfaction with the practicality of the procedures, while Bob’s mother suggested increasing the frequency and incorporating in-school implementation to enhance effectiveness. Participant interviews also indicated strong acceptance of the intervention. Both participants expressed positive attitudes and reported that the relaxation strategies supported their completion of assignments. Neither participant reported discomfort with the physiological monitoring procedures. Overall, these findings demonstrate high social validity, with caregivers and participants alike regarding the intervention as acceptable, feasible, and beneficial.

## 3. Results

### 3.1. Behavioral and Physiological Relaxation

As shown in Figure 1 and Figure 2, both participants demonstrated substantial improvements in performing the target relaxation behaviors; these were accompanied by physiological changes indicative of a relaxed state.

#### 3.1.1. Bob

During the pre-intervention probes, Bob’s behavioral relaxation ranged from 1% to 20% (M = 10%). Physiologically, his pulse rate ranged from 70 to 77 bpm (M = 72 bpm), and frontal EMG amplitude ranged from 4 to 13 μV (M = 10 μV), indicating a tense state.

Following the intervention, Bob’s behavioral relaxation increased substantially to 90–96% (M = 92%), his heart rate decreased to 61–69 bpm (M = 65 bpm), and EMG amplitude dropped to 2–4 μV (M = 3 μV), reflecting reduced muscle activity and an overall relaxed state. The consistent improvements across behavioral and physiological measures suggest that the relaxation training effectively promoted multidimensional relaxation for Bob.

#### 3.1.2. Sam

As shown in Figure 2, Sam’s behavioral relaxation prior to the intervention ranged from 1% to 8% (M = 6%). Physiological measures reflected a tense state: his heart rate ranged from 77 to 80 bpm (M = 78 bpm) and frontal EMG amplitude ranged from 12 to 22 μV (M = 16 μV), indicating elevated muscle activity and physiological arousal.

Following the intervention, Sam’s behavioral relaxation increased markedly to 91–94% (M = 92%), indicating successful acquisition of the target behaviors. His EMG amplitude decreased substantially to 3–4 μV (M = 3 μV), reflecting a physiologically relaxed state. Although his heart rate slightly increased to 79–81 bpm (M = 80 bpm), this range substantially overlapped with his baseline values, remained well below the threshold for physiological arousal (90 bpm), and therefore did not indicate physiological tension. It is also possible that minor fluctuations were influenced by factors unrelated to relaxation (e.g., sensor tightness during the evaluation). Because there was no meaningful difference between pre- and posttest readings, the change is likely attributable to natural variability rather than to the intervention itself. Taken together, the improvements in behavioral performance and reductions in muscle activity indicate that the intervention was effective in promoting multidimensional relaxation in Sam.

### 3.2. Academic Task Completion

Following the intervention, both participants showed significant and consistent increases in the percentage of completed task items across subjects. These improvements were largely maintained for one month (see Figure 3 and Figure 4).

#### 3.2.1. Bob

During baseline, Bob’s data were generally stable at low levels across subjects, though performance in English tasks showed greater variability than in Chinese and math (Chinese: 14–27%, M = 21%; math: 9–19%, M = 14%; English: 10–44%, M = 32%) as well as behaviors such as prolonged inactivity, scratching, nail-biting, off-task writing, and verbal expressions of frustration. Following the introduction of BRT, Bob showed immediate and substantial increases in academic task completion Chinese performance rose to a stable high level (range: 54–100%, M = 80%), math showed a clear increase in magnitude and trend (range: 38–93%, M = 69%), and English demonstrated a clear increasing trend and low within-phase variability (range: 67–100%, M = 92%). There was no overlap between baseline and intervention data for any of the subjects. Tau for nonoverlap with baseline trend control (Tau-U; [47]) showed significant phase differences in data across all subjects (Chinese: Tau-U = 1.06, *p* ≤ 0.001; math: Tau-U = 0.73, *p* = 0.040; English: Tau-U = 0.94, *p* = 0.010).

During the maintenance phase, Bob’s academic performance remained consistently high (Chinese: 96–100%, M = 99%; math: 92–100%, M = 97%; English: 100%). Bob’s mother reported that Bob occasionally initiated relaxation sessions independently before completing schoolwork, suggesting generalization of the intervention effects beyond the training context.

#### 3.2.2. Sam

During baseline, Sam’s academic task completion exhibited relatively low levels overall, accompanied by a downward trend across subjects, although performance was relatively high in the first session, likely due to few distractions in the learning environment. Specifically, task completion percentage in Chinese literacy declined steadily from 52% to 10% (M = 28%), and math followed a similar pattern, decreasing from 51% to 19% (M = 34%). For English tasks, completion initially declined and then showed a modest upward trend; nevertheless, data within the phase exhibited substantial variability (range: 0.3–66%, M = 22%). Instructors observed that Sam frequently engaged in self-stimulatory vocalizations and scribbling during baseline.

Following BRT, Sam’s academic performance improved significantly across all subjects. For Chinese literacy, task completion quickly increased and stabilized above criterion (range: 72–100%; M = 98%), with no overlap with baseline data and a significant change (Tau-U = 1.3, *p* = 0.010). Math completion also rose rapidly from 70% to 96% (M = 85%) and was maintained at a high level with occasional variation (Tau-U = 1.24, *p* = 0.010). Notably, Sam demonstrated subvocalization and increased task focus during task problem-solving. English task completion also improved significantly (range: 50–93%; M = 76%), with limited baseline overlap and a significant change (Tau-U = 1.01, *p* = 0.00). However, its variability during the intervention phase was notably greater than that observed in the other two subjects. It is worth noting that variability in data corresponded with observed tic behaviors and self-stimulatory actions.

During the maintenance phase, Sam’s improvements were largely sustained. Chinese task completion remained high (83–97%; M = 92%), and math stabilized at an even higher level than during intervention (87–100%; M = 96%). English performance also improved overall (82–100%; M = 90%). Sam’s mother reported that Sam continued practicing behavioral relaxation at home as part of his night routine, which may have contributed to his maintenance of treatment effects.

## 4. Discussion

The present study examined functional relations among BRT, behavioral and physiological indicators of relaxation, and academic performance as measured by task completion percentages. We observed positive effects of BRT in both participants across all three academic subjects. Relative to pre-intervention probes, participants, following BRT, demonstrated higher levels of behavioral relaxation as assessed through clinical observation and video coding, as well as increased physiological relaxation as indicated by reductions in pulse rate and frontal muscle EMG amplitude. Furthermore, both participants achieved significantly higher task completion percentages across three core academic subjects during the intervention phase and maintained these improvements throughout the maintenance phase.

Complementing prior studies with populations of different ages or diagnoses (e.g., [7]; [46]), our results demonstrate that BRT can also reduce anxiety and enhance academic performance in young students with autism. Unlike existing evidence on various complicated or high-technology-aided treatments for academic anxiety and engagement, this study highlights the value of evaluating a cost-effective intervention for students with autism in inclusive educational contexts. Specifically, compared with [10] ([10]), who tested a virtual reality biofeedback game, and [14] ([14]), who implemented a consumer-grade neurofeedback-assisted mindfulness-based intervention, the BRT treatment used in this study is simpler to understand, train, and apply across daily contexts. Moreover, different from studies that focused solely on behavioral indicators of task performance, this study incorporated both behavioral and physiological measures of anxiety, providing a potential avenue to explore the relation between mental health (anxiety) and academic performance, as well as the underlying mechanisms through which this specific treatment improves academic outcomes.

### 4.1. Why BRT Works for Students with Autism

To further understand the mechanisms underlying these outcomes, it is important to explore the possible reasons why students with autism achieved notable relaxation effects through BRT.

The findings suggest that BRT enhanced the students’ self-control and the frequency with which they applied relaxation strategies. According to [63] ([63]), self-control refers to controlling one part of one’s behavior in order to alter the probability of another behavior. For our participants, tension functioned as an aversive stimulus. Through repeatedly practicing the relaxation of specific body parts, they reduced or eliminated the occurrence of tension. Over time, these self-controlled relaxation behaviors were strengthened through automatic reinforcement and increased in frequency.

BRT supported the acquisition of relaxation behaviors because the intervention is grounded in applied behavior analysis and emphasizes observable, clearly defined behaviors. This approach enabled instructors to provide immediate feedback, reinforcing correct performance and correcting errors. The use of video modeling further facilitated learning by allowing students repeatedly to observe target behaviors, maintain attention, and quickly master relaxation of specific body parts or movements. Consequently, participants were able to adopt relaxation strategies effectively and accurately, improving their overall ability to perform the target relaxation behaviors.

In addition to behavioral improvements, participants achieved significant physiological relaxation effects. Post-intervention data showed a marked reduction in frontalis EMG amplitude, confirming physiological relaxation, which may be explained by evidence that muscle relaxation lowers EMG activity ([31]). Furthermore, based on the four-domain response theory of complex behavior, teaching overt relaxation behaviors through BRT elicited co-variation across other response domains. Specifically in the covert motor domain, EMG activity decreased, and in the overt verbal domain, students actively reported feeling “very relaxed” after training.

### 4.2. Why BRT Can Support Academic Performance in Students with Autism

The results indicate a functional relation between BRT and improvements in students’ task completion. Although further research and more specific data collection are needed to fully clarify the underlying mechanisms, several potential explanations may account for the effects of BRT.

Anecdotal information collected prior to the baseline phase and during social validity assessments suggested that BRT functioned as a form of natural reinforcement. As reported by teachers and caregivers, both participants frequently expressed a strong desire to relax each day, particularly before beginning academic tasks. By teaching BRT and encouraging its use before schoolwork, the intervention directly addressed these students’ needs for relaxation and self-regulation, leading to reduced pre-task anxiety and increased readiness to engage in learning activities. Because of this naturally reinforcing effect, participants became more likely to engage in behavioral relaxation. Based on within-session observations and participants’ and caregivers’ reports, these students could have experienced a calmer emotional state, maintained greater focus, and demonstrated improved persistence, all of which might have contributed to enhanced academic performance. Notably, both participants reported enjoying the intervention and expressed willingness to continue using behavioral relaxation techniques in the future.

While previous studies have demonstrated the effectiveness of BRT in reducing anxiety among both typically developing individuals and those with neurodiverse conditions ([65]; [21]; [46]), [11] ([11]) further identified a significantly negative relation between anxiety, motivation, and academic self-concept. Accordingly, reducing anxiety through BRT may have fostered improvements in students’ motivation and academic self-concept, as suggested, though not directly measured, by their verbal expressions and behaviors. Beyond behavioral and physiological indicators of anxiety, our participants, prior to receiving BRT, displayed explicit verbal expressions of negative affect when prompted to begin academic tasks (e.g., “Are you targeting me?” “Is there a lot of homework today?”). They also expressed low self-evaluations regarding their academic abilities (e.g., “This is too difficult for me to complete.”). Following the implementation of BRT, both participants demonstrated noticeable reductions in the frequency and intensity of negative emotions, restricted and repetitive behaviors (e.g., finger biting, head scratching), and challenging behaviors (e.g., tantrums) prior to academic activities. Concurrently, they exhibited increased motivation and more positive self-evaluations, often reporting greater confidence in their ability to complete assignments and even to complete more assignments than before.

Furthermore, the enhancement of attention may also have played a role in supporting task completion. Observations indicate that after receiving BRT, participants demonstrated longer periods of concentration and made more consistent efforts to complete their assignments, efforts such as rereading instructions aloud, performing calculations, and verifying their answers. Prior research has shown that attention problems and higher levels of depression are predictive of lower academic achievement in areas such as spelling and math computation ([39]). Therefore, the increased task completion observed in students with autism following BRT may be attributed to the intervention’s effectiveness in alleviating negative emotional experiences and mitigating attention difficulties.

An additional advantage of BRT for students with autism lies in its potential for seamless implementation across various settings. Caregivers of both participants reported that they applied BRT at home without direct involvement of clinicians or researchers, suggesting a preliminary efficacy. [48] ([48]) noted that environmental factors such as time constraints and limited resources can impact the generalization of interventions; they emphasized the importance of adapting programs so they can be delivered by teachers and caregivers across contexts without requiring extensive professional support. Moreover, the present BRT protocol incorporated clear, observable, and measurable teaching goals and strategies, required only 15 min per session, and utilized simple, low-cost materials (e.g., visual prompts, a pre-recorded instructional audio and/or video). These characteristics make BRT both time- and cost-efficient, supporting its feasibility for broader application within school settings to help improve the academic performance of students with autism.

### 4.3. Limitation

Several limitations of our study should be noted.

First, we did not assess the accuracy of our participants’ academic task completion. Instead, the study focused on instances in which participants experienced anxiety that hindered task completion or led them to complete only a small number of tasks. Previous meta-analyses have reported a very small effect size between higher anxiety levels and lower academic achievement as measured by grades ([11]; [16]). Therefore, the primary aim of this study was to examine the effect of BRT on increasing the quantity rather than the quality of academic tasks completed. Accordingly, data were collected solely on the number of Chinese, mathematics, and English task items completed within a 10-min period. Yet, for students with autism in inclusive schools, the quality or accuracy of academic task completion may be more educationally meaningful than the sheer amount of work completed. We acknowledge that the lack of accuracy data might have limited the comprehensiveness of the findings regarding overall academic performance. Future research should incorporate this dimension of measurement to provide a more complete understanding of the potential effectiveness of BRT.

Second, this study did not fully assess the generalization effects of the intervention; instead, we examined the one-month maintenance of improved task completion after the intervention. Even though caregivers were recommended to support the practice of BRT at home, no consistent data was collected because of family schedules and caregiver availability. Therefore, we did not have an opportunity to observe whether these effects extended to everyday learning environments such as school classes and home settings.

Third, we observed a slight increase in Sam’s pulse rate; however, there was no meaningful visual or numerical difference between pretest and posttest data, and neither stage exceeded the relaxation threshold that would indicate physiological tension. Because we lacked additional physiological indicators beyond frontal EMG amplitude, while the latter showed the more noticeable improvement that provided stronger evidence of BRT’s effect on physiological relaxation, we cannot fully explain the small fluctuation in pulse rate. It may have been influenced by factors such as sensor placement or tightness. Nonetheless, the limitation of relying on a potentially insufficient set of physiological measures is acknowledged, and future studies should incorporate a broader range of more precise assessment tools to address this issue.

Finally, the sample size and age range of our participants were relatively limited. Previous research suggests that younger students may experience a stronger impact of anxiety on academic achievement than do adolescents ([11]; [69]). Further research with larger and more diverse samples is therefore needed if we are to better understand the efficacy and effectiveness of BRT in supporting the development of students with autism across different ages.

### 4.4. Implications for Future Research and Practice

Future research might examine the effects of BRT on multiple dimensions of academic performance in students with autism. This study focused on percentage of task items completed, without assessing other aspects of performance. Future studies could investigate both the quality and duration of task completion. Integrating these indicators would allow a more comprehensive understanding of how BRT influences academic functioning in students with autism.

Second, future research could incorporate direct measures of academic anxiety, academic self-concept, and motivation. Collecting data in these areas may help clarify the mechanisms through which BRT influences the academic performance of students with autism. Even simple embedded questioning or self-report methods could be used to determine whether BRT effectively reduces task anxiety and enhances academic self-concept and motivation, thereby providing more direct evidence of its impact.

Third, it is important to examine the effectiveness of BRT across different educational contexts. In school settings, students with autism can find it difficult to follow teachers’ instructions and to complete classroom tasks and would benefit from additional support. Future studies could explore teacher-delivered BRT to help students achieve relaxation in class, potentially improving task engagement and performance.

Fourth, future research should extend the application of BRT to students with autism across different age groups. As students progress through middle school, high school, and university, they encounter increasing academic demands, greater social complexity, and significant physical and psychological changes. Investigating whether BRT can help alleviate academic stress and social anxiety at different developmental and social stages would provide valuable insights into its long-term benefits for both academic and social functioning.

## Figures and Tables

**Figure 1 behavsci-15-01633-f001:**
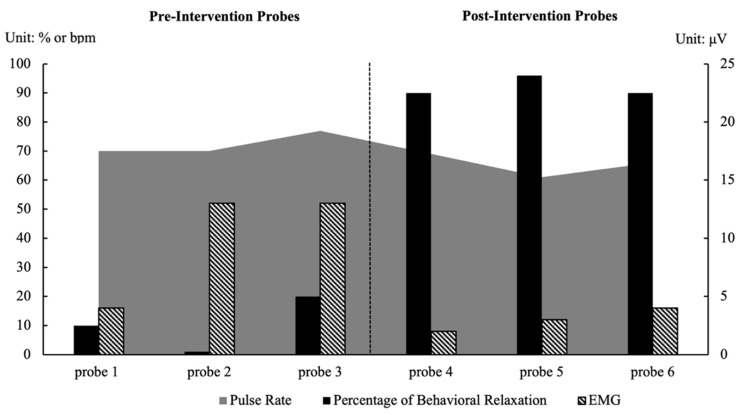
Results of Bob’s Behavioral and Physiological Relaxation.

**Figure 2 behavsci-15-01633-f002:**
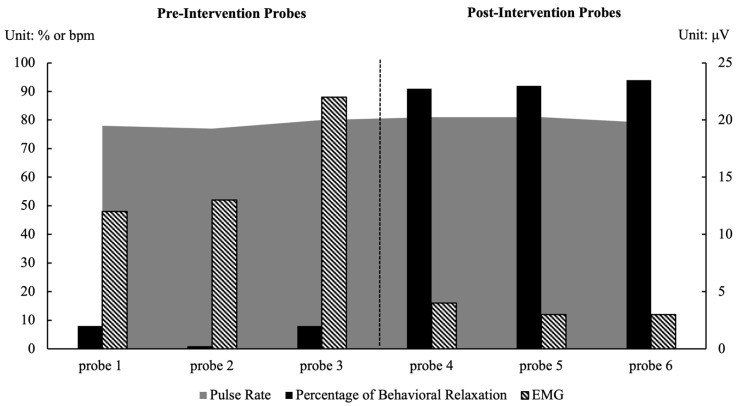
Results of Sam’s Behavioral and Physiological Relaxation.

**Figure 3 behavsci-15-01633-f003:**
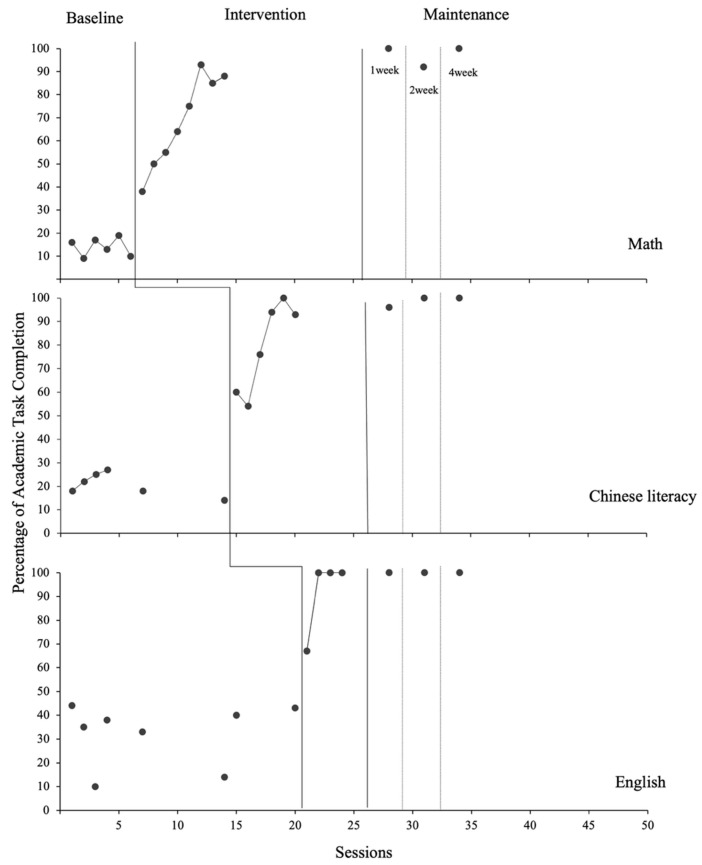
Results of Bob’s Academic Task Completion.

**Figure 4 behavsci-15-01633-f004:**
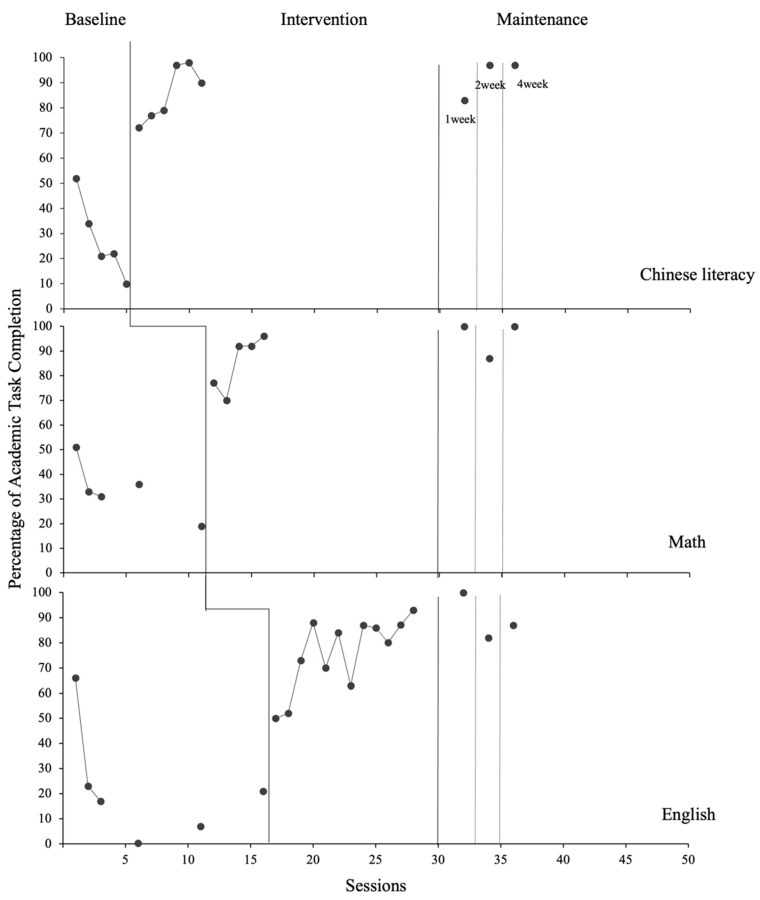
Results of Sam’s Academic Task Completion.

## Data Availability

The data that support the findings of this study are available from the corresponding author upon reasonable request.

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
