# Peer review of "The Effects of Behavioral Relaxation Training on Academic Task Completion Among Students with Autism in Inclusive Classrooms: A Single-Subject Design Study"

_behavsci, 2025, doi:10.3390/bs15121633_

Round 1

Reviewer 1 Report

Comments and Suggestions for Authors

Manuscript ID: behavsci-3990299

The study presents a behavioral relaxation training (BRT) to decrease the anxiety levels (tension as an aversive stimulus) in two students (Bob and Sam, aged 8 and 14 years respectively) in three academic subjects: Chinese literacy, math and English. Physiological, behavioral and academic performance were measured during the pre-intervention, intervention and the post-intervention (four weeks after intervention) phases. The results show important relaxation levels and increased percentages in the academic completion tasks in the three subjects assessed. Relaxation did work as a reward stimulus and a perception of behavior self-control for both students.

I find the manuscript easy to read and very interesting. Interventions in children with autism are necessary and the focus in only two students has proved the future usefulness of behavioral relaxation training in other students with autism. Caregivers’ and teachers’ perceptions about the generalization to other settings seem hopeful. This study is just an initial and promising step, wider studies with a higher number of participants are necessary in future research.

Title

It should indicate that this is a two cases study.

 Keywords and abstract

They are adequate.

Introduction

The introduction places the study in the broad context of people with autism. It includes a current and relevant review of the field with key publications cited. It clearly states the purpose and aims of the study, highlighting the very few works performed in children with autism using the behavioral relaxation training (BRT) to decrease the anxiety levels in school settings. Finally, three important research questions are added at the end of the section.

The introduction is well written and perfectly comprehensible for any reader.

Methods

Exclusion criteria should include the use of any kind of medication that could interfere with the treatment of relaxation. At the very least, include this issue in the limitations of the study.

In general, methods section is very complete, explaining in a very detailed way any aspect of the intervention. The baseline phase is very important when physiological assessment is used to avoid reactivity.

I only miss two aspects

  1. Identity of the caregiver, that is, was the father, the mother, the teacher? Please include this information (see 2.9 subsection, social validity)
  2. Information about the informed consent and ethical approval by the university ethical committee.

Results

Lines 402-403; 421-422, 426: authors should use three decimal places in the statistical “p” and avoid .00. If the significance is very low, for example, lower than .001, they can use the symbols “≤”. (≤ .001) 

Lines 431-433 authors again are commenting about caregivers, please, tell which role these caregivers have.

Discussion

Well developed, to the point and answering the research questions framing them in previous studies performed in different samples and settings.

I would not add anything else.

Authors should include information about the following aspects:

Author Contributions

Funding

Institutional Review Board Statement or Ethics Committee

Informed Consent Statement

Data Availability Statement

Acknowledgments

Conflicts of Interest.

References

The titles of articles are, in some cases, capitalized in each word but not in others. Please, capitalize only the first word of the article.

Author Response

Dear Reviewer:

Thank you for your thoughtful and constructive feedback on our manuscript. We appreciate your recognition of the relevance and potential of this study and have carefully addressed all your insightful suggestions. Below is a summary of our revisions in response to your comments:

Comment 1: It should indicate that this is a two cases study.

Response 1: We have now explicitly indicated that this study employs a single-subject design. To avoid potential confusion between single-subject designs and case study designs, we chose not to use the term “two-case study” and instead specify the single subject methodology applied.

Comment 2: Include medication use in exclusion criteria or limitations.

Response 2:We have added a sentence in the Participants section to clarify this issue:Children were excluded if they had previously received relaxation training, were currently participating in such services, or were taking medications likely to affect physiological arousal or relaxation response (e.g., anxiolytics, sedatives). (lines 204-205)

Comment 3: Clarify the identity of caregivers.

Response 3:We have specified the caregivers' identities,see lines 204-205.

Comment 4: Include ethical approval and informed consent.

Response 4:We have added the following statement in the Method section, see lines 205-206.

Comment 5: Use appropriate p-value formatting.

Response5 :We have corrected all p-values to three decimal places, avoiding the use of “.00”. Where appropriate, we now use “p ≤ .001” to indicate high significance.

Comment 6: Provide information on the research team.

Response 6:A new Research Team subsection has been added to the Methods section, summarizing the backgrounds of the interventionists and supervisory procedures, see lines 229-237.

Comment 7: Authors should include information about the following aspects(author contributions, funding, conflicts, etc.)

Response 7: Additional information has been added at the end of the paper, before References.

Comment 8: The titles of articles are, in some cases, capitalized in each word but not in others. Please, capitalize only the first word of the article.

Response 8: Necessary revisions have been made.

We appreciate your helpful guidance, which has significantly improved the quality of our manuscript. Please let us know if further clarification is required.

Sincerely,

On behalf of all co-authors

Reviewer 2 Report

Comments and Suggestions for Authors

Thank you for the opportunity to review the manuscript titled, The Effects of Behavioral Relaxation Training on Academic Task Completion Among Students with Autism in Inclusive Classrooms. The topic is relevant to inclusive education and the growing need for interventions that address both the behavioral and academic needs of autistic students. In the following sections, I presented feedback for each major section of the paper.

Introduction: Introduction provided overview of both academic and mental health challenges experienced by students with autism. Rationale for exploring BRT as an alternative to cognitively demanding anxiety interventions is articulated and the review of existing BRT research is provided. Research questions are included. However, the Introduction could be further strengthened by following revisions:

  • Theoretical argument linking anxiety, physiological arousal and academic task completion could be more explicitly connected to single-case intervention design literature and specific variables assessed in this study.
  • Limited articulation of gaps in BRT research specifically related to school-based application and younger children with autism in inclusive settings; this would improve justification for current study.
  • Introduction could explicitly describe why a multiple-probe across tasks design is appropriate for addressing the listed research questions.
  • Cultural context (Chinese inclusive schools) is not introduced until later; mentioning it briefly in the Introduction would strengthen contextual framing.

Methods: Methods section is organized well and provided details on participants, training procedures, measurement systems and fidelity checks. Description of the prerequisite skill training, BRT components and phase changes were included. High procedural fidelity and IOA values are reported. Despite these strengths, several methodological issues existed:

  • Only two participants were included which aligns with single-case methodology, but the Methods section does not justify why two participants were sufficient to demonstrate functional relations across three academic tasks each.
  • selection of tasks (previously mastered items) ensures stable responding, but the authors did not justify whether tasks were equated for difficulty or whether different tasks across sessions may have introduced variability.
  • If I understand it correctly, no data were collected on the accuracy of academic task completion which raises questions about whether BRT increased work production at the expense of correctness.
  • Decision rule requiring participants to meet a 90% relaxation accuracy before beginning task probes may unintentionally confound intervention effect with prerequisite skill training.
  • Generalization probes were not included.
  • More information is needed regarding qualifications of interventionist and training provided.
  • authors reference the four-domain response theory but do not operationally connect this framework to the dependent variables measured in the study.

Results: Results section presented clear functional relations between BRT and improvements in both relaxation and academic task completion. Behavioral and physiological measures are aligned and mutually supportive. Visual analysis supplemented by Tau-U effect size calculations provided strong argument for intervention impact across phases and participants. Nevertheless, several areas of the Results section could be improved:

  • Although the graphs are referenced, the narrative relies heavily on descriptive summaries; more explicit commentary on level, trend, variability and immediacy of effect would strengthen visual analysis.
  • Physiological data show mixed patterns (e.g., Sam’s slight heart rate increase), but the discussion does not fully address these discrepancies.
  • variability in English task completion for both participants deserves greater explanation and could indicate differential effects across academic domains.

Discussion: Discussion provided a nicely developed interpretation of why BRT may support academic engagement. authors connected their findings to existing research on anxiety and self-regulation in autism. implications for caregiver- and teacher-implemented BRT are relevant to inclusive classroom contexts. At the same time, the Discussion could be strengthened by the following revisions.

  • Some causal explanations (e.g., BRT improving motivation and academic self-concept) extend beyond the data collected; these should be framed more cautiously as hypotheses.
  • Discussion of mechanisms (e.g., four-domain response theory, self-control) is interesting but speculative without direct measurement.
  • limitation regarding lack of accuracy data is acknowledged but understated given its impact on educational significance.
  • Discussion did not compare results to prior single-case studies of anxiety reduction or academic engagement interventions.
  • Cultural considerations (e.g., Chinese inclusive education practices, parental expectations) are not integrated into interpreting results although they may influence anxiety and performance.
  • Some recommendations for future research (e.g., developing new anxiety scales) are important but could be more tightly linked to gaps revealed by the present study.

Overall, this manuscript addresses an important topic. The study demonstrates clear functional relations and provides valuable initial evidence for BRT as a feasible, low-cost intervention.

Author Response

Dear Reviewer:

Thank you for your thoughtful and constructive feedback on our manuscript. We appreciate your recognition of the relevance and potential of this study and have carefully addressed all your insightful suggestions. Below is a summary of our revisions in response to your comments.

Comment 1: Theoretical argument linking anxiety, physiological arousal and academic task completion could be more explicitly connected to single-case intervention design literature and specific variables assessed in this study.

Response 1: Thank you for the comment. Additional information has been provided in the section “1.1. Academic Performance and Academic Anxiety.” Specifically, we have added statements reflecting prior findings on the associations between academic anxiety, physiological arousal, and academic performance (lines 77-79). We also briefly summarized previous research using group designs and single-subject designs on this topic (lines 93-99). Furthermore, we clarified current research and practice gaps regarding treatments, physiological measures, and single-subject intervention evidence (lines 104-125), which underscores the significance of the present study.

Comment 2: Limited articulation of gaps in BRT research specifically related to school-based application and younger children with autism in inclusive settings; this would improve justification for current study.

Response 2: We have incorporated this specific research gap into our statement and further clarified the significance of the present study (lines 194-200).

Comment 3: Introduction could explicitly describe why a multiple-probe across tasks design is appropriate for addressing the listed research questions.

Response 3: We understand that including a brief description of the rationale can be helpful. However, we believe that different single-subject designs could be used to address these research questions, and our choice of a multiple-probe design was driven primarily by practical considerations. Rather than adding a paragraph in the Introduction specifically to justify the utility of the multiple-probe design, which might lead readers unfamiliar with single-subject methodology to think that this is the only appropriate approach, we have provided a more detailed explanation in Section 2.3, Research Design, under the Methods. This section clarifies the appropriateness of our chosen design (page 5, lines 248–255).

Comment 4: Cultural context (Chinese inclusive schools) is not introduced until later; mentioning it briefly in the Introduction would strengthen contextual framing.

Response 4: We have now introduced our cultural context earlier, at the end of the first paragraph (lines 48–52).

Comment 5: Some causal explanations (e.g., BRT improving motivation and academic self-concept) extend beyond the data collected; these should be framed more cautiously as hypotheses.

Response 5: We have tempered the statement and clarified that these inferences are based on behavioral observations rather than on quantitative data (page 14 lines 567-569).

Comment 6: Discussion of mechanisms (e.g., four-domain response theory, self-control) is interesting but speculative without direct measurement.

Response 6: Yes. We have now specified that on page 15 lines 583-584 (“as suggested, though not directly measured, by their verbal expressions and behaviors”) and also discussed it in Implications.

Comment 7: limitation regarding lack of accuracy data is acknowledged but understated given its impact on educational significance.

Response 7: We agree with this feedback and have added further discussion on the importance of task completion accuracy, highlighting its significance for consideration in future research. See page 16, lines 628-633.

Comment 8: Discussion did not compare results to prior single-case studies of anxiety reduction or academic engagement interventions.

Response 8: The comparison between the present study and prior evidence has been added as the second paragraph of the Discussion section. We highlighted both the alignment of our findings with previous research on the effectiveness of BRT training and the innovative aspects of the current study relative to earlier investigations (lines 514–527).

Comment 9: Cultural considerations (e.g., Chinese inclusive education practices, parental expectations) are not integrated into interpreting results although they may influence anxiety and performance.

Response 9: We acknowledge the importance of applying a cultural lens in scientific research. However, for this study, we believe it is more appropriate to interpret the findings cautiously within cultural contexts, both because of a) the limited relevance of certain cultural factors to our research questions and b) the lack of data collected on these dimensions.

Regarding the former, although we recognize that Chinese inclusive education practices and caregiver expectations for children’s academic performance may have unique characteristics, we do not see these factors as likely to influence the effectiveness of BRT on academic performance in students with autism. The measurement approaches used to assess anxiety, relaxation, and academic performance in this study are also not highly sensitive to cultural differences. For this reason, we are concerned that interpreting our results as culturally influenced may be confusing for international readers.

Regarding the latter, the BRT intervention implemented in this study was not designed to target national inclusive education practices or parental expectations. Thus, the improvements observed from baseline to intervention and maintenance are unlikely to be attributable to these factors. Moreover, because we did not collect data on these variables, we have no empirical basis for drawing conclusions about their potential influence on the outcomes.

To maintain a clear focus on the primary goal of the study, which is examining the potential benefits of BRT for young students with autism in inclusive settings, we have decided not to incorporate cultural interpretations into the discussion.

Comment 10: Some recommendations for future research (e.g., developing new anxiety scales) are important but could be more tightly linked to gaps revealed by the present study.

Response 10: Thank you for the feedback. We have revised the manuscript to recommend incorporating direct measurements of academic anxiety, self-concept, and motivation, which aligns with our earlier response regarding the speculative explanations of the underlying mechanisms (page 16, lines 639–643).

Comment 11: Physiological data show mixed patterns (e.g., Sam’s slight heart rate increase), but the discussion does not fully address these discrepancies.

Response 11: Thank you for highlighting this. we clarified that Sam’s heart rate remained well below the physiological tension threshold, and the slight increase likely reflected natural variability or sensor-related factors in the Results section (lines 466-472)ï¼›And we acknowledged the limitation of relying on a limited set of physiological indicators and suggested that future studies should include more comprehensive physiological assessments (lines 665-674).

Comment 12: Only two participants were included which aligns with single-case methodology, but the Methods section does not justify why two participants were sufficient to demonstrate functional relations across three academic tasks each.

Response 12: Thank you for this observation. We have revised the Methods section to clarify that although only two participants were included, the use of a concurrent multiple-probe design across three academic tasks per participant allowed repeated demonstrations of treatment effects across distinct behaviors, supporting the identification of functional relations. (lines 242–248)

Comment 13:Selection of tasks (previously mastered items) ensures stable responding, but the authors did not justify whether tasks were equated for difficulty or whether different tasks across sessions may have introduced variability.

Response 13: Thank you for the insightful suggestion. We have added an explanation on how academic worksheets were selected and standardized to ensure task equivalence and stable difficulty across sessions (lines 280–288).

Comment 14:If I understand it correctly, no data were collected on the accuracy of academic task completion which raises questions about whether BRT increased work production at the expense of correctness.

Response 14: We have added clarification to indicate that task accuracy was not recorded, as the study focused specifically on behavioral engagement and sustained task participation, not academic correctness(lines 277-279). Further discussion on this limitation and its implications for educational significance has also been provided in the Discussion section.

Comment 15:Decision rule requiring participants to meet a 90% relaxation accuracy before beginning task probes may unintentionally confound intervention effect with prerequisite skill training.

Response 15: Thank you for your insightful comment. 90% relaxation accuracy (as measured by the BRS) was used not to train a prerequisite skill, but to ensure that participants entered the academic task in a clear, observable relaxed state. This criterion enables us to examine whether overt relaxation behaviors (as taught through BRT) also elicited physiological relaxation and improved academic engagement. Rather than a confounding factor, the BRS criterion was integral to the intended treatment process.

Comment 16:Generalization probes were not included.

Response 16: Thank you for the comment. This issue has been acknowledged in the Limitations section (lines 665-670). We noted that generalization data were not collected due to constraints related to family schedules and caregiver availability, and we discussed the need for future research to examine generalization effects in naturalistic contexts such as home and classroom environments.

Comment 17:More information is needed regarding qualifications of interventionist and training provided.

Response 17: Thank you for the helpful suggestion. We have added details to clarify the qualifications of the interventionists and the training process they received prior to the intervention. see lines 229-237.

Comment 18:Authors reference the four-domain response theory but do not operationally connect this framework to the dependent variables measured in the study.

Response 18: Thank you for this valuable suggestion. We have now explicitly connected the four-domain response theory to the dependent variables in the Methods section. Specifically, we clarified how behavioral relaxation (lines 303-305, 315-316,321-328).

Comment 19:Although the graphs are referenced, the narrative relies heavily on descriptive summaries; more explicit commentary on level, trend, variability and immediacy of effect would strengthen visual analysis.

Response 19: We have revised the Results section to provide more detailed visual analysis for each participant and subject(lines 490-492,497-500,511-512,516-518).

Comment 20:Variability in English task completion for both participants deserves greater explanation and could indicate differential effects across academic domains.

Response 20: We have addressed this comment by expanding our narrative analysis in the Results section to describe the observed variability in English task performance (lines 529-531)

We appreciate your helpful guidance, which has significantly improved the quality of our manuscript. Please let us know if further clarification is required.

Sincerely,

On behalf of all co-authors

Round 2

Reviewer 2 Report

Comments and Suggestions for Authors

Thank you to authors for carefully addressing my feedback across all sections of the manuscript. The revised version of The Effects of Behavioral Relaxation Training on Academic Task Completion Among Students with Autism in Inclusive Classrooms: A Single-Subject Design Study shows clear attention to suggestions provided in previous round of review. I am satisfied with the current version the paper.